# Phenotypic and Molecular Characterization of Multidrug-Resistant Clinical Isolates of the *Candidozyma haemuli* Species Complex (Formerly *Candida haemulonii* Species Complex) from the Brazilian Amazon Reveals the First Case of *Candidozyma pseudohaemuli* in Brazil

**DOI:** 10.3390/jof11050394

**Published:** 2025-05-20

**Authors:** Sérgio Lobato França, Rodrigo Santos de Oliveira, Gabriel Silas Marinho Sousa, Sarah Rodrigues de Sá, Walber da Silva Nogueira, Elaine Patrícia Tavares do Espírito Santo, Daniel dos Santos Caldas, Silvia Helena Marques da Silva

**Affiliations:** 1Hospital Universitário João de Barros Barreto, Belém 66073-000, Pará, Brazil; walbernogueira@gmail.com; 2Programa de Pós-Graduação em Biologia de Agentes Infecciosos e Parasitários, Instituto de Ciências Biológicas, Universidade Federal do Pará, Belém 66075-110, Pará, Brazil; gmarinho676@gmail.com (G.S.M.S.); dancaldas@yahoo.com (D.d.S.C.); 3Laboratório de Micoses Superficiais e Sistêmicas, Seção de Bacteriologia e Micologia, Instituto Evandro Chagas, Ananindeua 67030-000, Pará, Brazil; rodrigodeoliveira01@gmail.com (R.S.d.O.); sarah.rdsa@aluno.uepa.br (S.R.d.S.); elainetavares@iec.gov.br (E.P.T.d.E.S.); silviasilva@iec.gov.br (S.H.M.d.S.)

**Keywords:** nosocomial infections, base sequence, virulence factors, antifungal agents, Brazil

## Abstract

This study included 24 isolates of the *Candidozyma haemuli* species complex from patients in the Brazilian Amazon between 2021 and 2024. These isolates were identified by sequencing as *C. duobushaemuli* (54.2%), *C. haemuli* sensu stricto (29.2%), *C. haemuli* var. *vulneris* (12.5%), and *C. pseudohaemuli* (4.2%). The finding of *C. pseudohaemuli* represents the first case reported in Brazil. Haplotype and phylogenetic analysis of these species, along with other isolates from Brazil, revealed low intraspecific genetic diversity. Resistance to at least one antifungal was observed in 83.3% of isolates, with multidrug resistance in 58.3%, including one isolate resistant to all tested antifungals. The isolates demonstrated active biofilm production, lytic enzyme activity, and thermotolerance. Notably, one *C. duobushaemuli* isolate exhibited tolerance to 42 °C, a phenotype not previously described. It is crucial for Brazil and other countries to recognize the emergence of these species as a public health threat and to take proactive measures to prevent outbreaks.

## 1. Introduction

Recently, a new genus called *Candidozyma* has been proposed to aggregate the *Candidozyma haemuli* species complex (CHSC) (formerly known as the *Candida haemulonii* species complex), together with the multidrug-resistant fungus *Candidozyma auris* (formerly *Candida auris*) [1].

This new genus includes both species with significant clinical relevance and those isolated from natural environments, indicating a broad ecological and adaptive diversity [1]. These fungi are notable not only for their resistance to antifungal drugs but also for their ability to cause invasive fungal infections, which can be difficult to treat and have high mortality rates, especially in immunocompromised patients [2,3].

Although the global prevalence of the CHSC is still relatively low compared to other fungal infections, the number of cases has increased worryingly in recent years [2,3,4,5,6]. It is interesting to note that in South America, infections with this complex are more frequently associated with candidemia, while in Asia and the United States, cases of cutaneous and non-invasive infections predominate [7,8].

Brazil has stood out as one of the countries most affected by this increase in cases of resistant fungal infections, with the number of diagnoses almost doubling over the last 11 years [9]. This reflects both the growing resistance of fungi to conventional treatments and the lack of early diagnosis of these infections [5].

In contrast to *C. auris*, which is a focus of intensive global study and monitoring, knowledge about the CHSC is still limited. This includes not only an understanding of its biology but also the paucity of epidemiological data on its geographical distribution and the factors that contribute to its pathogenicity [10,11].

Recent studies have shown that, in particular, clinical isolates of the CHSC in the Brazilian Amazon region remain virtually unexplored [4,10]. Considering that the distribution and frequency of infections caused by different species of the complex may vary according to geographical region [12], it is possible that ecological and structural factors specific to the Amazon may influence the circulation and dissemination of these species.

Therefore, the main objective of this study was to identify and characterize clinical isolates of the *Candidozyma haemuli* species complex from patients in the Brazilian Amazon region in order to increase understanding of the presence, diversity, virulence, and antifungal susceptibility of these species in the region.

## 2. Materials and Methods

### 2.1. This Study

This is a cross-sectional observational study that included CHSC isolates obtained from patients treated at a referral hospital in the state of Pará, Brazilian Amazon, from January 2021 to May 2024. Sociodemographic and clinical information of the patients was collected by reviewing medical records.

### 2.2. Species Identification

CHSC isolates were initially identified using the Vitek^®^ 2 compact system (BioMérieux, Durham, NC, USA) and the MALDI-TOF Biotyper^®^ sirius (Bruker, Billerica, MA, USA), following the manufacturers’ recommendations. Final species identification, phylogenetic analyses, and haplotype studies were conducted through genetic sequencing of the ITS1-5.8S-ITS2 rDNA regions using the ABI 3130 genetic analyzer (Thermo Fisher Scientific, Waltham, MA, USA). Sequences were aligned with those deposited in the GenBank database (NCBI, https://www.ncbi.nlm.nih.gov, accessed on 12 October 2024) referencing type strains, with species determination based on a similarity and coverage threshold of >99%. Phylogenetic and haplotype analyses incorporated sequences from this study along with non-redundant Brazilian sequences available in GenBank.

Sequences were first organized into a multifasta file using NotePad++ v8.5.4 (Don HO, https://notepad-plus-plus.org, accessed on 5 October 2024) and then aligned using MAFFT v7 (RIMD, https://mafft.cbrc.jp/alignment/server, accessed on 9 October 2024). Phylogenetic tree reconstruction was performed using IQ-TREE (http://www.iqtree.org, accessed on 7 October 2024), applying the maximum likelihood method with the TNe+G4 substitution model and 1000 bootstrap replicates. Haplotype analysis included the evaluation of polymorphic diversity using DnaSP v5.10 (http://www.ub.edu/dnasp/index_v5.html, accessed on 10 October 2024) and haplotype network construction was done using Cytoscape v3.10.2 (https://cytoscape.org, accessed on 14 October 2024).

### 2.3. Antifungal Susceptibility Testing

Antifungal susceptibility testing was conducted for amphotericin B (AB), fluconazole (FLU), itraconazole (ITC), and flucytosine (FC) (all from Sigma-Aldrich, Burlington, MA, USA) following the European Committee on Antimicrobial Susceptibility Testing (EUCAST) broth microdilution protocol E.DEF.7.4 [13]. Since clinical susceptibility breakpoints for CHSC are not established, we used provisional thresholds for *C. auris* suggested by the Centers for Disease Control and Prevention (CDC) [14] for FLU (≥32 µg/mL) and AB (≥2 µg/mL). For ITR (≥1 µg/mL) and FC (≥32 µg/mL), the general *Candida* spp. breakpoints from the Clinical and Laboratory Standards Institute (CLSI) were applied (M27-A3) [15].

### 2.4. Evaluation of Virulence Factors

Virulence factors evaluated included thermotolerance at 30 °C, 37 °C, and 42 °C on Sabouraud dextrose agar plates after 48 h of incubation, biofilm formation, biomass quantification, and enzymatic activities of lipase, phospholipase, and proteinase.

#### 2.4.1. Biofilm Formation and Biomass Quantification

The assays for biofilm formation, biomass quantification, and metabolic activity assessment were performed as described by Ramos et al. [16] and in technical triplicate. For this, fungal cell suspensions were prepared in Sabouraud broth, and 200 µL of each suspension (containing 10^6^ cells) were transferred into the wells of a 96-well microplate and incubated at 37 °C for 48 h without agitation. After the incubation period, the supernatant was carefully removed, and the wells were washed with PBS to eliminate non-adherent cells. The biofilms were then fixed with 99% methanol and allowed to air dry.

To quantify the biomass, the biofilms were stained with 0.4% crystal violet solution, washed with PBS, and subsequently decolorized with 33% acetic acid. The absorbance of the resulting solution was measured at 560 nm using a microplate reader to estimate the amount of biomass present in the biofilms.

Finally, the metabolic activity of the biofilms was assessed using the XTT/menadione colorimetric assay. After preparation of the XTT/menadione solution and its addition to the wells containing the biofilms, the plates were incubated at 37 °C for 3 h in the dark. The supernatant was then transferred to a new 96-well microplate, and absorbance was measured at 492 nm, enabling the quantification of the fungal biofilms’ metabolic activity.

To assess differences in biomass among species, the non-parametric Kruskal–Wallis test was applied due to unequal sample sizes and non-normally distributed data.

#### 2.4.2. Phospholipase Activity

Phospholipase activity was assessed following the method described by Price et al. [17]. Sabouraud agar supplemented with 8% egg yolk, 1 M NaCl, and 5 mM CaCl_2_ was used to prepare the phospholipase agar medium. The test samples were initially cultured on Sabouraud agar for 48 h. Subsequently, colonies were inoculated onto phospholipase agar at equidistant points and incubated at 37 °C for 10 days. The diameters of the colonies with precipitation zones were measured according to the method described by Price et al. [17].

#### 2.4.3. Lipase Activity

Lipase activity was evaluated according to the method described by Muhsin et al. [18]. The lipase agar medium was prepared by supplementing a base medium with the following components: 10 g of peptone (Merck, Darmstadt, HE, Germany), 5 g of NaCl (VETEC, Brazil), 0.1 g of CaCl_2_ (Sigma-Aldrich, Burlington, MA, USA), 20 g of agar (Merck, Darmstadt, HE, Germany), and 10 mL of Tween 20 (Sigma-Aldrich, Burlington, MA, USA). A portion of each colony was inoculated onto sterile Petri dishes containing the lipase agar and incubated at 27 °C for 10 days. Samples were considered lipase producers when an opaque halo was observed around the colony.

#### 2.4.4. Proteinase Activity

Proteinase activity was assessed according to the method described by Rüchel et al. [19]. The proteinase agar medium was prepared by supplementing the base medium with the following components: 11.7 g of Yeast Carbon Base (HIMEDIA, Mumbai, MH, India), 2 g of bovine serum albumin—fraction V (Sigma-Aldrich, Burlington, MA, USA), 2.5 mL of Protovit^®^ (Roche, Basel, Switzerland), and 100 mL of sterile distilled water. Samples were inoculated onto the medium and incubated at 30 °C for 10 days. Proteinase production was indicated by the formation of a clear halo around the colony.

All enzymatic activity assays were performed in technical duplicate and quantified using the activity index (Pz) [17].

### 2.5. Statistical Analysis

ANOVA was employed to analyze biofilm results, while the Kruskal–Wallis test followed by the Dwass–Steel–Critchlow–Fligner post hoc test was used for enzymatic activity data. Pearson’s correlation coefficient was used for the correlation analysis. Statistical analyses were performed using JAMOVI software (v2.3, https://www.jamovi.org, accessed on 15 November 2024), with statistical significance set at *p* < 0.05.

### 2.6. Ethics Approval

This study was approved by the research ethics committee for human studies of the João de Barros Barreto University Hospital (No. 5.514.367) and the Evandro Chagas Institute (No. 5.799.031) in compliance with Brazilian legislation.

## 3. Results

### 3.1. Patient Characteristics and Isolate Identification

We analyzed 24 clinical CHSC isolates from 24 patients (Table 1). A comparison of identification methods, using sequencing as the reference standard, revealed discrepancies in results obtained with Vitek^®^ 2 (66.7%) and MALDI-TOF (13.3%) (Table 2).

Phylogenetic reconstruction showed no genetic divergence among Brazilian isolates (Figure 1). A similar result was observed in haplotype diversity analysis. Although two haplotypes were identified for *C. pseudohaemuli*, this separation was considered insignificant as it resulted from a single mutation (Figure 2). Notably, no *C. pseudohaemuli* sequences from Brazil were available in GenBank.

### 3.2. Phenotypic Virulence Features of the Isolates

All isolates were able to grow at 35 °C and 37 °C; however, at 42 °C, only one isolate of *C. duobushaemuli* (IEC-CAND248) exhibited growth (Figure 3).

Regarding biofilm formation, all isolates demonstrated production capacity, and there was a positive correlation between biofilm biomass and metabolic activity (Pearson’s r = 0.547) (Figure 4). The analysis of mean biomass among species, conducted using the Kruskal–Wallis test, did not indicate a statistically significant difference between the evaluated groups (H = 1.27; *p* = 0.736). This suggests that, under the assay conditions, the analyzed species exhibited similar performance in terms of biomass production. The results of the quantification of biofilm biomass and metabolic activity in clinical isolates of the *Candidozyma haemuli* species complex and reference strains are presented in Appendix A.

In enzymatic activity tests, isolates showed varying capacities to produce proteinase (*p* = 0.017) and lipase (*p* = 0.005). Pairwise analysis for proteinase revealed differences between *C. duobushaemuli* and *C. haemuli* sensu stricto (*p* = 0.015), primarily due to 85.7% of *C. haemuli* sensu stricto isolates being strong producers of this enzyme, while *C. duobushaemuli* isolates were either moderate producers (53.8%) or non-producers (30.8%). For lipase, differences were observed between *C. duobushaemuli* and *C. haemuli* sensu stricto (*p* = 0.004), as well as between *C. duobushaemuli and C. haemuli* var. *vulneris* (*p* = 0.013). These differences were mainly attributed to the fact that none of the *C. duobushaemuli* isolates produced lipase (Table 3).

### 3.3. Antifungal Resistance Patterns in Clinical Isolates

Antifungal susceptibility testing revealed that most isolates displayed some level of resistance (83.3%). Multidrug resistance (MDR) was observed in half of the isolates, a common finding for *C. duobushaemuli* (69.2%) and *C. pseudohaemuli* (100%) (Table 4 and Table 5). Multidrug-resistant isolates were at least resistant to azoles and amphotericin B (AB), with the following resistance combinations identified: FLU+ITC+AB (20.8%); FLU+AB (16.7%); ITC+AB (8.3%); FLU+ITC (8.3%); and FLU+ITC+AB+FC (4.2%).

## 4. Discussion

This is the first study of clinical isolates of CHSC from the Brazilian Amazon, highlighting the discovery of *C. pseudohaemuli*, which represents the first report of this species in Brazil, as there were no previous records [9,20,21,22,23,24,25,26,27,28,29,30]. This report represents a significant advance in the understanding of clinical fungal diversity in the country. The absence of previous reports suggests that this species may have been underdiagnosed due to limitations in conventional identification methods or that its emergence is recent in the region.

Furthermore, the identification of *C. pseudohaemuli* in the Brazilian Amazon highlights the urgency of an effective surveillance system for monitoring emerging fungal infections. Epidemiological surveillance contributes to the early detection of new cases, the identification of dissemination patterns, and the implementation of appropriate control measures, especially in hospital environments, where species of the *Candidozyma* genus can cause outbreaks that are difficult to manage due to their recognized resistance to antifungal drugs [31,32].

With regard to *C. pseudohaemuli*, only a slightly over a dozen cases have been reported worldwide to date, with cases identified in South Korea, Thailand, the United States, Venezuela, Panama, Colombia, and French Guiana, involving fungemias and antifungal resistance profiles [30,33,34,35]. The limited distribution of *C. pseudohaemuli* positions it as one of the rarest pathogens in the field of CHSC. The detection of this fungus in the Amazon underscores the importance of microbiological surveillance in the region, especially given the growing emergence of opportunistic pathogens in tropical environments [30].

Genetic analysis of the rDNA regions of the isolates in this study, including other Brazilian isolates, revealed low intraspecific diversity within CHSC in Brazil, consistent with a Brazilian study that used whole genome sequencing [29].

This low diversity suggests a possible clonal expansion of these species in the country [36], which could have important implications for public health, facilitating the spread of well-adapted strains and increasing the risk of outbreaks. The stability of genotype frequencies may be associated with the small number of sequenced isolates, low mutation rates, or possible clonal origins [37], highlighting the need for continuous genetic surveillance to monitor evolutionary dynamics and implement more effective control strategies.

In terms of virulence, in addition to tolerance at 35–37 °C, one isolate of *C. duobushaemuli* showed growth at 42 °C, which is, to our knowledge, the first global report of this phenotype for the species. Previous studies have not demonstrated this characteristic [6,38,39,40,41,42,43]. Until now, it was only known that *C. auris*, *C. khanbhai,* and *C. ruelliae*, within the genus *Candidozyma*, could withstand 42 °C [6,44,45]. However, genomic studies have shown that *C. duobushaemuli* and other CHSC species possess genes involved in thermotolerance [10].

This characteristic could increase the virulence and evolutionary advantage of *C. duobushaemuli*, allowing it to survive in hosts during febrile episodes and in high-temperature environments. Although more studies are needed, we infer that local environmental conditions and the environmental adaptability of CHSC suggest that this high thermotolerance may represent an adaptation of *C. duobushaemuli* to rising temperatures in the Amazon [46,47,48].

We also observed the ability of the isolates to form biofilms. Biofilm formation by *Candidozyma* spp. is particularly relevant due to its association with increased antifungal resistance, evasion of host immune responses, and adaptation to stress conditions, which complicate treatment and may promote infection persistence [16].

Regarding enzymatic activity, our analysis revealed notable differences among the species evaluated. *C. haemuli* and its varieties exhibited higher levels of proteinase and lipase activity compared to *C. duobushaemuli* isolates. These findings are consistent with previous studies that also reported elevated enzymatic activity in *C. haemuli* [16,26,27,49], suggesting a possible role for these enzymes in virulence and tissue invasion. For *C. duobushaemuli*, the absence of lipase production in the isolates may represent a local pattern, although a Brazilian isolate with very high lipase activity has been described [44]. Phospholipase activity was low in all isolates of this study, in agreement with the results of other authors [23,26,27,49], which may suggest a less significant role for this enzyme in the pathogenicity of these isolates. However, further studies are needed to better investigate this aspect.

The low prevalence of CHSC and the limited studies on its phenotypic characteristics may be related to the difficulty of laboratory identification using commercial methods. In our comparative analysis of identification methods, frequent failures of Vitek^®^ 2 were observed, which is consistent with the results of Ambaraghassi et al. [50]. Although MALDI-TOF performed better, discrepancies in results were also observed, which highlights the need to expand mass spectrometry databases for CHSC identification. Given the limitations of conventional identification methods, we believe that many CHSC infections in hospitals in Brazil and around the world are underreported.

Clinically, the main challenge of CHSC infections in the Brazilian Amazon is antifungal resistance, particularly due to the high number of isolates resistant to azoles and AB, a common feature in *C. duobushaemuli* and *C. pseudohaemuli.* This resistance limits therapeutic options, making the management of these infections more complex, especially in immunocompromised patients or those with associated comorbidities [11,13]. However, in the present study, a favorable observation was the broad susceptibility of these isolates to FC, encouraging its use as a therapeutic alternative.

Despite the potential of FC, its isolated use has limitations, since monotherapy can lead to the rapid development of resistance and has variable efficacy [51]. It is therefore recommended that FC be administered in combination with other antifungals to ensure a synergistic effect and broaden the therapeutic response, reducing the fungal load and minimizing the likelihood of emerging resistance [52].

The resistance profile of the *C. pseudohaemuli* isolate (IEC-CAND49) to FLU and AB was similar to that of the first recorded isolate [33] and other global isolates [30], potentially indicating an intrinsic resistance to these antifungals. A worrying finding was the extensive resistance exhibited by an isolate of *C. duobushaemuli* to three classes of antifungals. Locally, this isolate can be considered the true “superfungus”, since all the strains of *C. auris* identified in Brazil to date have not shown resistance [53,54,55]. Regarding the country’s therapeutic capacity to combat these fungi, there is cause for concern, as a recent survey indicated that, in 2019, half of Brazilian hospitals had no echinocandins and less than 20% had FC available [56]. In this context, the absence of echinocandins in our testing is recognized as a limitation, which occurred due to constraints within our study.

In conclusion, the Brazilian Amazon emerges as a region with a diverse presence of CHSC species in hospital environments, characterized by frequent multidrug resistance and isolates exhibiting distinct virulence traits, such as elevated thermotolerance, which pose a potential global health threat. The socio-economic profile of the region, characterized by limited access to healthcare and inadequacies in hospital infrastructure, may facilitate the dissemination of these pathogens and complicate their timely diagnosis and treatment. Our findings also highlight the relevance of considering both environmental and hospital-associated transmission routes in the epidemiology of CHSC infections. In this context, the implementation of regional genomic surveillance programs becomes essential to monitor strain circulation, detect emerging variants, and inform public health strategies. These results reinforce the need for continuous monitoring efforts in the region to enable early detection of CHSC cases, guide therapeutic decision-making, and mitigate the burden of infection on vulnerable populations. To effectively limit CHSC infections, hospitals must strengthen their diagnostic capacity and ensure equitable access to antifungal agents.

## Figures and Tables

**Figure 1 jof-11-00394-f001:**
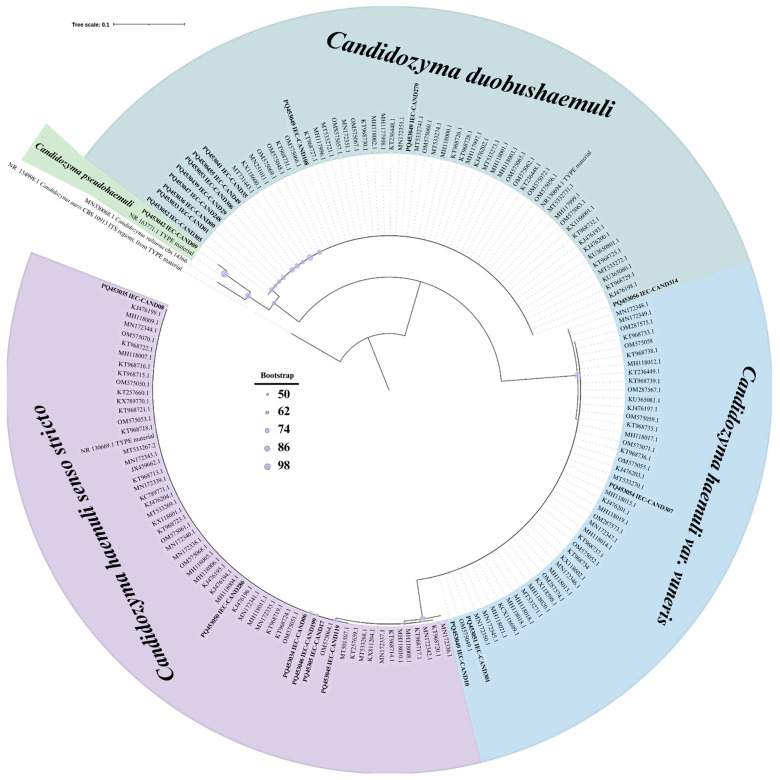
Maximum likelihood phylogenetic tree of CHSC isolates from Brazil based on ITS rDNA region sequences. The tree was constructed by clustering ITS region sequences from the Amazon region (n = 24) with other Brazilian sequences (n = 140) available in GenBank (https://www.ncbi.nlm.nih.gov, accessed on 12 October 2024), using 1000 bootstrap replicates. Sequences highlighted in bold represent isolates from the Amazon region. A strain of *Candida albicans* (CBS 562) was used as the outgroup. Maximum likelihood bootstrap values are displayed as circles above the nodes, with circle size proportional to the bootstrap value.

**Figure 2 jof-11-00394-f002:**
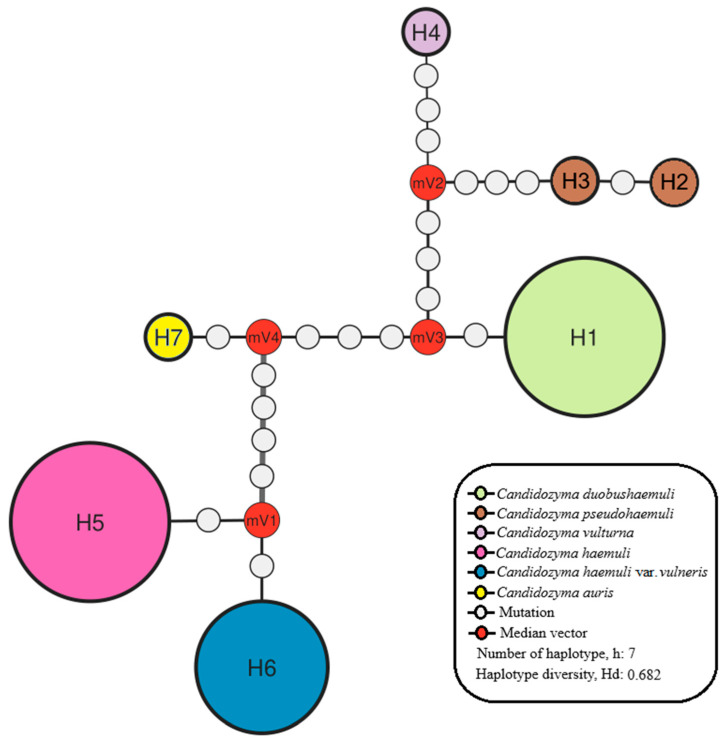
Haplotype diversity of CHSC isolates from Brazil based on ITS rDNA region sequences. The haplotype network was generated by clustering ITS region sequences from the Amazon region (n = 24) with other Brazilian sequences (n = 140) available in GenBank (https://www.ncbi.nlm.nih.gov, accessed on 12 October 2024). Each circle represents a haplotype coded by color (H1–H7). The size of each circle is proportional to the haplotype frequency. Each line in the network represents a mutational step. Red dots (median vectors, mv) indicate haplotypes not displayed or extinct in the population.

**Figure 3 jof-11-00394-f003:**
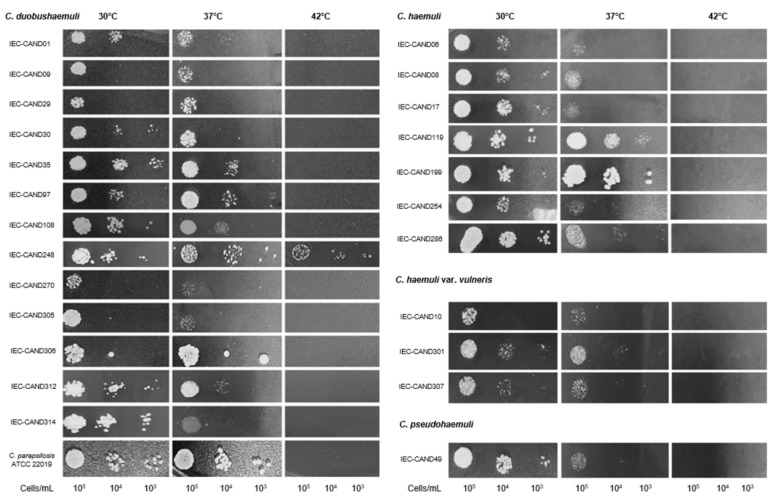
Thermotolerance assessment of clinical isolates of the CHSC, Brazilian Amazon, 2021–2024. The isolates were cultured on Sabouraud Dextrose Agar plates at different dilutions and temperatures for 48 h. *Candida parapsilosis* ATCC 22019 was used as the control strain.

**Figure 4 jof-11-00394-f004:**
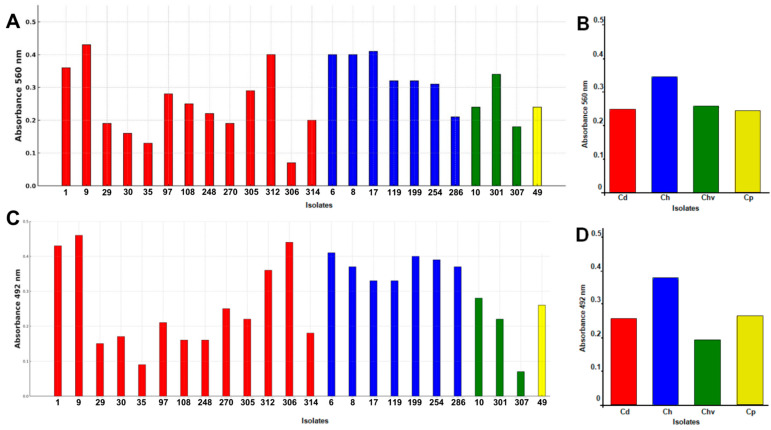
Biofilm formation by clinical isolates of the CHSC, Brazilian Amazon, 2021–2024. Quantification of biofilm biomass was performed for each isolate, stained with crystal violet and measured by optical density (560 nm) after 48 h at 37 °C (**A**). The mean biofilm biomass quantification for each fungal species is presented (**B**). The viability of biofilm-associated cells was assessed using the XTT(2,3-bis(2-methoxy-4-nitro-5-sulfophenyl)-5-[(phenylamino)carbonyl]-2H-tetrazolium hydroxide) assay and quantified by optical density (492 nm) after 3 h of incubation at 37 °C (**C**). The mean results of cell viability tests for biofilm-associated cells are shown by fungal species (**D**). Species abbreviations: Cd (red), *C. duobushaemuli*; Ch (blue), *C. haemuli*; Chv (green)*, C. haemuli* var. *vulneris*; and Cp (yellow), *C. pseudohaemuli*.

**Table 1 jof-11-00394-t001:** Characteristics of patients with CHSC yeast isolation, Brazilian Amazon, 2021–2024 *.

Variables	No. (%)
Demographic variables	
Age range	
13–18	1 (4.2)
19–30	3 (12.5)
31–45	5 (20.8)
46–60	7 (29.2)
61–75	4 (16.7)
76+	4 (16.7)
Male gender	16 (66.6)
Female gender	8 (33.3)
Sites of isolation	
Bronchoalveolar lavage	14 (58.3)
Sputum	4 (16.7)
Tracheal secretion	2 (8.3)
Urine	1 (4.2)
Skin lesion	1 (4.2)
Blood	1 (4.2)
Pleural fluid	1 (4.2)
Main clinical condition	
Tuberculosis	5 (20.8)
Chronic pneumopathy †	5 (20.8)
Neoplasia	5 (20.8)
PLHIV + tuberculosis	3 (12.5)
PLHIV + cryptococcosis	1 (4.2)
PLHIV + histoplasmosis	1 (4.2)
Erysipelas	2 (8.3)
Paracoccidioidomycosis	1 (4.2)
Pneumonia	1 (4.2)
Clinical history	
Hospitalization	15 (62.5)
Intensive care	5 (20.8)
Prior antifungal therapy (within 60 days)	8 (33.3)
Prior antibiotic therapy (within 60 days)	17 (70.83)
Isolated species	
*C. duobushaemuli *	13 (54.2)
*C. haemuli* sensu stricto	7 (29.2)
*C. haemuli* var. *vulneris*	3 (12.5)
*C. pseudohaemuli *	1 (4.2)

* The total number of patients evaluated was 24. † Chronic pneumopathy included any chronic structural, interstitial, or obstructive pulmonary disease. PLHIV, people living with human immunodeficiency virus.

**Table 2 jof-11-00394-t002:** Identification of clinical isolates of the CHSC using different methodologies, Brazilian Amazon, 2021–2024 *.

Isolate No.	Source	Year	Age/ Gender	Identification			
				Previous Identification (VITEK^®^ 2) **	MALDI-TOF MS (Biotyper^®^ Sirius System) **	ITS Region Sequencing	GenBank Accession No.
IEC-CAND01	BAL	2021	46/M	* C. duobushaemulonii *	NA	* C. duobushaemuli *	PQ453033
IEC-CAND09	BAL	2021	33/M	* C. famata *	* C. duobushaemulonii *	* C. duobushaemuli *	PQ453036
IEC-CAND29	Tracheal secretion	2021	84/F	NA	* C. duobushaemulonii *	* C. duobushaemuli *	PQ453039
IEC-CAND30	Urine	2021	60/F	* C. famata *	* C. duobushaemulonii *	* C. duobushaemuli *	PQ453040
IEC-CAND35	BAL	2021	31/M	NA	* C. duobushaemulonii *	* C. duobushaemuli *	PQ453041
IEC-CAND97	Blood	2020	77/M	* C. duobushaemulonii *	* C. duobushaemulonii *	* C. duobushaemuli *	PQ453043
IEC-CAND108	BAL	2021	26/M	* C. duobushaemulonii *	* C. duobushaemulonii *	* C. duobushaemuli *	PQ453044
IEC-CAND248	BAL	2022	43/M	NA	* C. duobushaemulonii *	* C. duobushaemuli *	PQ453047
IEC-CAND270	Sputum	2022	65/F	NA	* C. duobushaemulonii *	* C. duobushaemuli *	PQ453049
IEC-CAND305	BAL	2022	82/F	* C. duobushaemulonii *	* C. duobushaemulonii *	* C. duobushaemuli *	PQ453052
IEC-CAND306	BAL	2022	57/M	* C. spherica/Saccharomyces cerevisiae * †	NA	* C. duobushaemuli *	PQ453053
IEC-CAND312	BAL	2022	40/M	* C. auris/ * * C. duobushaemulonii * †	* C. duobushaemulonii *	* C. duobushaemuli *	PQ453055
IEC-CAND314	BAL	2024	18/M	* C. haemulonii *	NA	* C. duobushaemuli *	PQ453056
IEC-CAND06	BAL	2021	30/M	* C. haemulonii *	NA	* C. haemuli *	PQ453034
IEC-CAND08	Tracheal secretion	2021	73/F	* C. parapsilosis *	NA	* C. haemuli *	PQ453035
IEC-CAND17	Sputum	2021	20/M	NA	* C. haemulonii *	* C. haemuli *	PQ453038
IEC-CAND119	BAL	2021	72/F	NA	* C. haemulonii *	* C. haemuli *	PQ453045
IEC-CAND199	Sputum	2021	42/M	NA	NA	* C. haemuli *	PQ453046
IEC-CAND254	Skin lesion	2022	51/F	* C. duobushaemulonii *	NA	* C. haemuli *	PQ453048
IEC-CAND286	sputum	2022	53/M	NA	* C. haemulonii *	* C. haemuli *	PQ453050
IEC-CAND10	BAL	2021	72/M	* C. famata *	* C. duobushaemulonii *	*C. haemuli* var. *vulneris*	PQ453037
IEC-CAND301	BAL	2022	49/M	*C. haemulonii/C. haemulonii* var. *vulneris* †	*C. haemulonii* var. *vulneris*	*C. haemuli* var. *vulneris*	PQ453051
IEC-CAND307	Pleural fluid	2022	50/F	*C. haemulonii* var. *vulneris*	*C. haemulonii* var. *vulneris*	*C. haemuli* var. *vulneris*	PQ453054
IEC-CAND49	BAL	2021	81/M	NA	* C. haemulonii *	* C. pseudohaemuli *	PQ453042

* The total number of patients evaluated was 24. ** Identifications performed by the VITEK^®^ 2 and MALDI-TOF MS systems followed the previous taxonomy of the *Candida* genus, in which the isolates were classified as part of the *Candida haemulonii* species complex. † The system was unable to differentiate between the indicated species. BAL, bronchoalveolar lavage; NA, not available.

**Table 3 jof-11-00394-t003:** Production of extracellular hydrolytic enzymes by isolates of the CHSC (n = 24) from patients in the Brazilian Amazon, 2021–2024 *.

Isolates	Proteinase			Lipase			Phospholipase		
	Pz Index	Activity	Median Pz (IQR)	Pz Index	Activity	Median Pz (IQR)	Pz Index	Activity	Median Pz (IQR)
*C. duobushaemuli* (*n* = 13)									
IEC-CAND09	0.69	Strong	0.82 (0.78–1.00) †	1.00	None	1.0 (1.0–1.10) †	1.00	None	1.0 (1.0–1.10)
IEC-CAND30	0.79	Moderate		1.00	None		1.00	None	
IEC-CAND01	0.75	Moderate		1.00	None		1.00	None	
IEC-CAND248	0.82	Moderate		1.00	None		1.00	None	
IEC-CAND270	0.73	Moderate		1.00	None		1.00	None	
IEC-CAND305	0.82	Moderate		1.00	None		1.00	None	
IEC-CAND312	0.78	Moderate		1.00	None		1.00	None	
IEC-CAND314	0.78	Moderate		1.00	None		1.00	None	
IEC-CAND29	0.90	Weak		1.00	None		0.85	Moderate	
IEC-CAND35	1.00	None		1.00	None		1.00	None	
IEC-CAND97	1.00	None		1.00	None		1.00	None	
IEC-CAND108	1.00	None		1.00	None		1.00	None	
IEC-CAND306	1.00	None		1.00	None		1.00	None	
*C. haemuli* (*n* = 7)									
IEC-CAND06	0.52	Strong	0.57 (0.54–0.61) †	0.53	Strong	0.57 (0.55–0.79) †	1.00	None	1.0 (1.0–1.10)
IEC-CAND17	0.57	Strong		0.56	Strong		1.00	None	
IEC-CAND119	0.55	Strong		0.57	Strong		1.00	None	
IEC-CAND286	0.50	Strong		0.37	Strong		1.00	None	
IEC-CAND08	0.65	Strong		1.00	None		1.00	None	
IEC-CAND254	0.57	Strong		1.00	None		0.83	Moderate	
IEC-CAND199	0.89	Moderate		0.57	Strong		1.00	None	
*Candida haemuloni* var. *vulneris* (*n* = 3)									
IEC-CAND301	0.54	Strong	0.75 (0.65–0.79) †	0.39	Strong	0.51 (0.45–0.76) †	1.00	None	1.0 (1.0–1.10)
IEC-CAND307	0.75	Moderate		0.51	Strong		1.00	None	
IEC-CAND10	0.83	Moderate		1.00	None		1.00	None	
*C. pseudohaemuli* (*n* = 1)									
IEC-CAND49	0.69	Strong	0.69 (0.69–0.69) †	1.00	None	1.0 (1.0–1.10) †	1.00	None	1.0 (1.0–1.10)

* Pz index, enzymatic activity zone; IQR, interquartile range. The ranges of activity according to the Pz index were established as follows: high, Pz ≤ 0.69; moderate, Pz = 0.70–0.89; weak, Pz = 0.90–0.99; none, Pz = 1. † Significant differences (*p* < 0.05), Kruskal–Wallis test.

**Table 4 jof-11-00394-t004:** Antifungal susceptibility test results for CHSC isolates (n = 24) from patients in the Brazilian Amazon, 2021–2024 *.

	MIC, μg/mL			
Clinical Isolates	FLU	ITC	AB	FC
*C. duobushaemuli* (*n* = 13)				
IEC-CAND01	>64	0.5	8	0.12
IEC-CAND09	4	>8	2	0.12
IEC-CAND29	16	0.5	1	0.12
IEC-CAND30	64	>8	≥8	0.25
IEC-CAND35	16	>8	1	0.25
IEC-CAND97	64	1	8	>64
IEC-CAND108	>64	8	>8	0.25
IEC-CAND248	64	0.5	4	0.25
IEC-CAND270	>64	>8	4	0.12
IEC-CAND305	>64	>8	>8	0.25
IEC-CAND306	>64	8	0.5	0.25
IEC-CAND312	4	0.25	>8	0.25
IEC-CAND314	>64	8	>8	0.12
*C. haemuli* (*n* = 7)				
IEC-CAND06	64	0.25	>8	0.12
IEC-CAND08	2	0.25	2	0.12
IEC-CAND17	8	0.25	>8	0.12
IEC-CAND119	2	0.5	1	0.12
IEC-CAND199	2	0.125	1	0.12
IEC-CAND254	64	1	1	0.12
IEC-CAND286	4	0.25	1	0.12
*C. haemuli* var*. vulneris* (*n* = 3)				
IEC-CAND10	2	1	1	0.12
IEC-CAND301	1	0.125	2	0.25
IEC-CAND307	16	2	2	0.12
*C. pseudohaemuli* (*n* = 1)				
IEC-CAND49	64	0.5	2	0.25

* Minimum inhibitory concentration (MIC) assays were performed following the E.Def 7.4 protocol of the European Committee on Antimicrobial Susceptibility Testing (EUCAST) (https://www.eucast.org/astoffungi/methodsinantifungalsusceptibilitytesting/susceptibility_testing_of_yeasts, 4 September 2024). MIC breakpoints were interpreted according to provisional criteria from the Centers for Disease Control and Prevention (CDC) for *Candidozyma auris* for fluconazole (≥32 µg/mL) and amphotericin B (≥2 µg/mL) (https://www.cdc.gov/candida-auris/hcp/laboratories/antifungal-susceptibility-testing.html, accessed on 7 September 2024). For itraconazole (≥1 µg/mL) and flucytosine (≥32 µg/mL), the M27-S3 protocol of the Clinical and Laboratory Standards Institute (CLSI) was followed. FLU, fluconazole; ITC, itraconazole; AB, amphotericin B; FC, flucytosine; MIC, minimum inhibitory concentration.

**Table 5 jof-11-00394-t005:** Percentage of resistant CHSC isolates (n = 24) from patients in the Brazilian Amazon, 2021–2024 *.

	% Resistance				
Clinical Isolates	FLU	ITC	AB	FC	%MDR
*C. duobushaemuli* (*n* = 13)	69.2	69.2	69.2	7.7	69.2
*C. haemuli* sensu stricto (*n* = 7)	28.6	14.3	42.9	0.0	14.3
*C. haemuli* var. *vulneris* (*n* = 3)	0.0	66.7	66.7	0.0	33.3
*C. pseudohaemuli* (*n* = 1)	100.0	0.0	100.0	0.0	100.0
Total (*n* = 24)	50.0	50.0	66.7	4.2	50.0

* Resistance percentages were calculated based on the minimum inhibitory concentration (MIC) values presented in Table 4. The breakpoints used to define resistance were fluconazole (≥32 µg/mL) and amphotericin B (≥2 µg/mL), according to provisional criteria established by the Centers for Disease Control and Prevention (CDC) for *Candidozyma auris*; itraconazole (≥1 µg/mL) and flucytosine (≥32 µg/mL), according to the M27-S3 protocol of the Clinical and Laboratory Standards Institute (CLSI). MDR (multidrug resistance) was defined as resistance to two or more classes of antifungal agents. FLU, fluconazole; ITC, itraconazole; AB, amphotericin B; FC, flucytosine; MDR, multidrug resistance.

## Data Availability

The original contributions presented in this study are included in the article and Appendix A. Further inquiries can be directed to the corresponding author.

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
