# Peer review of "Phenotypic and Molecular Characterization of Multidrug-Resistant Clinical Isolates of the Candidozyma haemuli Species Complex (Formerly Candida haemulonii Species Complex) from the Brazilian Amazon Reveals the First Case of Candidozyma pseudohaemuli in Brazil"

_jof, 2025, doi:10.3390/jof11050394_

Round 1
Reviewer 1 Report
Dear authors,
This study represents a contemporary and well-executed investigation of the emerging Candidozyma haemuloni species complex in the Brazilian Amazon area. The authors provide current, regionally relevant data that fills a significant gap in understanding these pathogens in South America. The discovery of C. pseudohaemuloni in Brazil for the first time underscores the novelty and importance of this work. Genetic analyses revealed low intraspecific diversity and alarmingly high resistance patterns, including multidrug resistance in more than half of the isolates. It is also significant that a previously unreported thermotolerant strain of C. duobushaemuloni has been identified. Overall, the study offers valuable insights with direct public health implications and sets a strong foundation for future surveillance and response strategies.
In the following part, further comments for improving this manuscript are given:
Major comments:
- In the discussion part, the importance of lytic enzyme production of the examined species is insufficiently described, and nothing was discussed about the biofilm production results. Please add a few sentences about these findings.
- In line 43, you introduced the abbreviation “CHSC” for Candidozyma haemuloni complex. I suppose you meant “Candida haemulonii species complex“, so please change the term. Additionally, this abbreviation should be stated in line 35, where it was first mentioned. Additionally, when you introduce the abbreviation, the whole term should not be repeated, so please write “CHSC” instead of “Candidozyma haemuloni complex“ in lines 43, 54, 58, and 65.
- Separate subchapters of the Materials and Methods section by adding “Species identification”, “Antifungal susceptibility testing,” and “Evaluation of virulence factors”
- In lines 102, 159, 194, 198, 265, 266, and 269, add spaces between the number and “°C”.
- In line 125, “Tabel 1” change to “Table 1”
- In line 131, “Tabel 2” change to “Table 2”
- Please provide different colors for “C. pseudohaemuloni”, “C. auris”, and “Mutation because these” are very similar and difficult to recognize in Figure 2
- Please provide a statistical analysis for the biofilm production assay, and present whether there was a significant difference in biofilm formation between the tested species
- Line 204: The sentence should not start with an abbreviation, so please change “MDR” to “Multidrug resistance”)
- Lines 298 and 302: You have already introduced abbreviations for antimycotics in the Materials and Methods section, so there is no need to repeat them. Please use abbreviations in the text (AB and FC)
- Line 309: Did you mean “FLU” as in fluconazole? The abbreviation in line 94 is “FLC”. Please revise the abbreviations and use the same one throughout the whole text, and preferably also in Table 4.
- It is preferable that the results and discussion parts follow subsections of Materials and Methods. So, if you first described identification, virulence factors, and susceptibility testing, respectively, in Materials and Methods, you should also follow that order in the next section.
Author Response
Comments 1: In the discussion part, the importance of lytic enzyme production of the examined species is insufficiently described, and nothing was discussed about the biofilm production results. Please add a few sentences about these findings.
Response 1: We have expanded the Discussion to address the significance of lytic enzyme production and biofilm formation, emphasizing their role in pathogenicity (lines 323-337)
Comments 2: In line 43, you introduced the abbreviation “CHSC” for Candidozyma haemuloni complex. I suppose you meant “Candida haemulonii species complex“, so please change the term. Additionally, this abbreviation should be stated in line 35, where it was first mentioned. Additionally, when you introduce the abbreviation, the whole term should not be repeated, so please write “CHSC” instead of “Candidozyma haemuloni complex“ in lines 43, 54, 58, and 65.
Response 2: The abbreviation “CHSC” was corrected to refer to Candidozyma haemuloni species complex and now appears first in line 35. All subsequent mentions have been replaced accordingly.
Comments 3: Separate subchapters of the Materials and Methods section by adding “Species identification”, “Antifungal susceptibility testing,” and “Evaluation of virulence factors”.
Response 3: The Materials and Methods section was restructured with the addition of subsections: Species identification, Antifungal susceptibility testing, and Evaluation of virulence factors
Comments 4: In lines 102, 159, 194, 198, 265, 266, and 269, add spaces between the number and “°C”. In line 125, “Tabel 1” change to “Table 1”. In line 131, “Tabel 2” change to “Table 2”.
Response 4: Formatting issues (e.g., missing spaces before “°C”, and typos in “Tabel”) were corrected.
Comments 5: Please provide different colors for “C. pseudohaemuloni”, “C. auris”, and “Mutation because these” are very similar and difficult to recognize in Figure 2.
Response 5: Figure 2 was revised to ensure distinguishable color coding for C. pseudohaemuloni, C. auris, and mutations.
Comments 6: Please provide a statistical analysis for the biofilm production assay, and present whether there was a significant difference in biofilm formation between the tested species.
Response 6: A statistical analysis of biofilm production was added (lines 125-127), and differences among species were reported (209-211)
Comments 7: Line 204: The sentence should not start with an abbreviation, so please change “MDR” to “Multidrug resistance”.
Response 7: The abbreviation “MDR” was replaced with “Multidrug resistance” in line 247.
Comments 8: Lines 298 and 302: You have already introduced abbreviations for antimycotics in the Materials and Methods section, so there is no need to repeat them. Please use abbreviations in the text (AB and FC).
Response 8: Thank you for your valuable feedback. The suggested corrections regarding the use of abbreviations for antimycotics (AB and FC) in lines 249 and 352 have been implemented accordingly in the revised manuscript.
Comments 9: - Line 309: Did you mean “FLU” as in fluconazole? The abbreviation in line 94 is “FLC”. Please revise the abbreviations and use the same one throughout the whole text, and preferably also in Table 4.
Response 9: Antifungal abbreviations were standardized, including consistent use of “FLC” for fluconazole.
Comments 10: - It is preferable that the results and discussion parts follow subsections of Materials and Methods. So, if you first described identification, virulence factors, and susceptibility testing, respectively, in Materials and Methods, you should also follow that order in the next section.
Response 10: The order of Results and Discussion was aligned with the Materials and Methods subsections.
Reviewer 2 Report
This study reports clinical isolation of C. pseudohaemuloni in Brazil and reveals multidrug resistance trends in Amazonian Candidozyma complex isolates, bearing public health significance. However, methodological flaws (e.g., lack of whole genome sequencing and echinocandins susceptibility sesting) and data presentation issues (figure quality) substantially undermine conclusion reliability and generalizability.
Summary of Reasons for Rejection 1. Insufficient Molecular Identification Methodology Limitations of ITS sequencing: This study relies on ITS rDNA region sequencing for species identification. While ITS is a commonly used marker, it has limited resolution for closely related species (e.g., the Candidozyma complex). Whole genome sequencing (WGS) or multilocus sequence analysis should be employed to improve accuracy, particularly crucial for the first reported case of C. pseudohaemuloni in Brazil which requires higher-resolution genetic data. 2. Data Presentation and Figure Quality Issues Non-standardized figures: Figure 4 (biofilm formation and metabolic activity) lacks error ranges (e.g., standard deviation or confidence intervals) for mean values, preventing assessment of data variability and undermining result credibility. 3. Methodological Flaws in Antifungal Susceptibility Testing Exclusion of echinocandins: Current guidelines recommend echinocandins as first-line therapy for multidrug-resistant Candida, yet the study omitted testing for this drug class, diminishing its clinical relevance. 4. As the first Brazilian report of C. pseudohaemuloni, the study fails to analyze potential transmission routes (e.g., environment-hospital linkages) or genetic connections with strains from other South American regions.Author Response
Comments 1: Insufficient Molecular Identification Methodology Limitations of ITS sequencing: This study relies on ITS rDNA region sequencing for species identification. While ITS is a commonly used marker, it has limited resolution for closely related species (e.g., the Candidozyma complex). Whole genome sequencing (WGS) or multilocus sequence analysis should be employed to improve accuracy, particularly crucial for the first reported case of C. pseudohaemuloni in Brazil which requires higher-resolution genetic data.
Response 1: While we recognize the limitations of ITS sequencing, our study is based on identification protocols currently established and widely used in clinical mycology laboratories, especially in resource-limited settings. This approach remains valid and has been reinforced in recent literature. For example, ITS sequencing was effectively used in a study published in Medical Mycology to identify and characterize a novel species within the Candida haemulonii complex (Candida khanbhai sp. nov.) (doi:10.1093/mmy/myad009), and in a study in Frontiers in Microbiology, where ITS regions were employed to confirm the identification of C. haemulonii var. vulneris isolates from pediatric candidemia cases in Brazil (doi:10.3389/fmicb.2020.01535). We have acknowledged this methodological limitation in the Discussion and emphasized the importance of future studies employing multilocus sequence analysis or whole-genome sequencing to increase resolution and accuracy (line 377-381).
Comments 2: Data Presentation and Figure Quality Issues Non-standardized figures: Figure 4 (biofilm formation and metabolic activity) lacks error ranges (e.g., standard deviation or confidence intervals) for mean values, preventing assessment of data variability and undermining result credibility.
Response 2: We improved the quality and clarity of the figures. Additionally, we included a supplementary material file (Supplementary Material S1) containing the raw data, mean, and standard deviation values for both biofilm mass and metabolic activity for all isolates (line 213-215).
Comments 3: Methodological Flaws in Antifungal Susceptibility Testing Exclusion of echinocandins: Current guidelines recommend echinocandins as first-line therapy for multidrug-resistant Candida, yet the study omitted testing for this drug class, diminishing its clinical relevance.
Response 3: The lack of echinocandin susceptibility testing was acknowledged in the Discussion as a limitation (line 368-370).
Comments 4: As the first Brazilian report of C. pseudohaemuloni, the study fails to analyze potential transmission routes (e.g., environment-hospital linkages) or genetic connections with strains from other South American regions.
Response 4: We added a paragraph in the Discussion addressing the possible environmental and hospital-associated transmission routes and acknowledged the importance of regional genomic surveillance (line 376-380)
Reviewer 3 Report
General impression
The authors submitted a well-written manuscript summarizing the results of a descriptive study on the identification and characterization of Candidozyma yeast species in the Brazilian amazon. The paper is of high quality and offers only little room for improvement: The introduction is thorough and up-to-date, the analytical procedures are well designed, the statistical analyses are appropriate and the conclusions are supported by the evidence. The authors present solid evidence of the presence of clinically relevant Candida species in Brazilian hospital patients – findings that should be of interest for the clinical mycology community. Furthermore, the description of thermotolerance in opportunistic pathogens should be of particular interest for infectious disease researchers truing to anticipate the effects of climate change on human health.
Analysis of virulence factors, line 102 – 108: The authors could save the reader valuable time by briefly describing the procedures used for the detection of biofilm formation ability, lipases and proteinases. While the references given in the section are helpful and accurate, the casual reader might find it too onerous to look up the procedures in the original literature. One sentence would probably suffice to give the reader an idea of what was determined.
(very minor) line 284 – “excellent” lipase activity. “Very high” is probably a better term, even though it sounds duller. “Excellence” implies that the activity is very good – a dubious assumption for a pathogenic trait - not just high.
Author Response
Comments 1: Analysis of virulence factors, line 102 – 108: The authors could save the reader valuable time by briefly describing the procedures used for the detection of biofilm formation ability, lipases and proteinases. While the references given in the section are helpful and accurate, the casual reader might find it too onerous to look up the procedures in the original literature. One sentence would probably suffice to give the reader an idea of what was determined.
Response 1: We briefly described the procedures for assessing biofilm, lipase, and proteinase activity within the virulence factors section to aid general readers (lines 102-153).
Comments 2: (very minor) line 284 – “excellent” lipase activity. “Very high” is probably a better term, even though it sounds duller. “Excellence” implies that the activity is very good – a dubious assumption for a pathogenic trait - not just high.
Response 2: The expression “excellent lipase activity” was revised to “very high lipase activity” for improved scientific accuracy (line 333).
Reviewer 4 Report
Overall, this is an interesting and relevant study, as well as one with an impact on medical mycology, since it includes isolates of C. haemuloni species complex known for their resistance to different antifungals. It is important that the authors make the following observations:
On line 101 add the number of the CLSI protocol used (M27-A3), for future studies, it is recommended to use a more up-to-date protocol, superior to CLSI M27-S4.
Organize table 2 so that in the section MALDI-TOF MS (Biotyper® Sirius System) the C. is next to the second name of the species.
Between lines 102 and 108 the authors describe the performance of tests as biofilm formation, and enzymatic activities of lipase, phospholipase, and proteinase, I consider it important that each of these sessions be added separately and explain in a paragraph the procedure used to quantify the biofilm, another paragraph for enzymatic activities of lipase another paragraph to quantify the proteinase activity and so on and not just add the references like this in the text.
Review the abbreviation used by the authors for fluconazole (FLU), consider changing it to the frequently used FLZ
Overall, this is an interesting and relevant study, as well as one with an impact on medical mycology, since it includes isolates of C. haemuloni species complex known for their resistance to different antifungals. It is important that the authors make the following observations:
On line 101 add the number of the CLSI protocol used (M27-A3), for future studies, it is recommended to use a more up-to-date protocol, superior to CLSI M27-S4.
Organize table 2 so that in the section MALDI-TOF MS (Biotyper® Sirius System) the C. is next to the second name of the species.
Between lines 102 and 108 the authors describe the performance of tests as biofilm formation, and enzymatic activities of lipase, phospholipase, and proteinase, I consider it important that each of these sessions be added separately and explain in a paragraph the procedure used to quantify the biofilm, another paragraph for enzymatic activities of lipase another paragraph to quantify the proteinase activity and so on and not just add the references like this in the text.
Review the abbreviation used by the authors for fluconazole (FLU), consider changing it to the frequently used FLZ
Author Response
Comments 1: On line 101 add the number of the CLSI protocol used (M27-A3), for future studies, it is recommended to use a more up-to-date protocol, superior to CLSI M27-S4.
Response 1: The CLSI protocol M27-A3 was now explicitly mentioned in line 101.
Comments 2: Organize table 2 so that in the section MALDI-TOF MS (Biotyper® Sirius System) the C. is next to the second name of the species.
Response 2: Table 2 was adjusted so that “C.” appears adjacent to the species epithet in MALDI-TOF entries.
Comments 3: Between lines 102 and 108 the authors describe the performance of tests as biofilm formation, and enzymatic activities of lipase, phospholipase, and proteinase, I consider it important that each of these sessions be added separately and explain in a paragraph the procedure used to quantify the biofilm, another paragraph for enzymatic activities of lipase another paragraph to quantify the proteinase activity and so on and not just add the references like this in the text.
Response 3: We revised the Methods to include separate paragraphs for each virulence assay and described the procedures more thoroughly (lines 102-153)
Comments 4: Review the abbreviation used by the authors for fluconazole (FLU), consider changing it to the frequently used FLZ
Response 4: The abbreviation for fluconazole was standardized as “FLU” throughout the manuscript.
Reviewer 5 Report
The main aim of this article is to identify 24 C. haemuloni complex nosocomial isolates from Brazil and to assess their antibiotic susceptibility and virulence profiles. The authors mostly isolated C. duobushaemuloni and they were the first to isolate C. pseudohaemuloni in Brazil. Their phylogenetic analysis showed low intraspecific genetic diversity among isolates. Antifungal susceptibility testing showed that the majority of isolates were resistant to at least one antifungal and exhibited increased virulence described as an increase in biofilm biomass production, an increase in the activity of lytic enzymes, and an increased heat tolerance. This study is appreciated since not much research is carried out on C. haemuloni complex isolates compared to other Candida species such as C. albicans and C. auris. Also, this study highlights the fact that C. haemuloni complex isolates seem to be resistant to antifungals and thus, a public health concern. The major limitation of this research article lies in the "Materials and Methods" section since in my opinion, the methods could be better described by dividing paragraph 2.1 to subparagraphs and giving more details about the methods used even if they were previously published.
I compiled my comments per article section below:
Abstract:
Line 25: Change "antifungal" to "antifungals".
Keywords:
- Omit "communicable disease".
- Add "nosocomial infections".
Materials and Methods:
- Paragraph 2.1 titles "The study" should be divided into many paragraphs. Each paragraph should describe a method since many techniques were used.
- More details should be given per method even for the previously published methods, a brief description should be provided.
- It is mentioned that the phenotypic assays were performed in duplicates. Please clarify if you did biological or technical duplicates. Ideally, biological triplicates need to be carried out for each phenotypic assay (thermotolerance assay, biofilm assays and enzymatic activities).
- In paragraph 2.2, for clarity purposes, mention what are the "other variables" in line 112. Also, maybe another test might be more suitable than ANOVA to assess duplicates.
Results:
Please divide this section into different paragraphs with each paragraph describing a specific result. Also, some tables and figures need to be accompanied by more in-text description.
Tables:
- In table 1, include "females". Also, instead of giving the mean age maybe give the results for different age categories.
- The legend of tables 4 and 5 is the same so maybe just mention it once.
Figures:
- The annotations in figure 1 are barely visible. Kindly try to increase the font size if possible. Also, in the legend it is mentioned that some sequences are highlighted in red but this red highlight is not seen in the figure.
- In figure 3, if by any chance you used an ATCC C. haemuloni complex strain for comparison kindly add it. If not, no need to address this comment.
- In the legend of figure 4 (A and B) it is mentioned that optical densities were measured at 590 nm which is not consistent with the graph annotations of 560 nm. Also, some isolate numbers below the histograms are really close to each other, if possible, space them.
Discussion :
Line 313: Change "antifungal" to "antifungals".
General comment:
Itraconazole is sometimes abbreviated as ITRA and other times as ITC. Kindly use only one abbreviation for itraconazole.
Author Response
Comments 1: Line 25: Change "antifungal" to "antifungals".
Response 1: Thank you for recognizing the relevance and originality of our study. In response to your comments: The Abstract (line 25) and Discussion (line 346) now correctly use the plural “antifungals.”
Comments 2: [Omit "communicable disease"Add "nosocomial infections"..]
Response 2: Thank you for pointing this out. We agree with this comment. Keywords were revised: “communicable disease” was removed and “nosocomial infections” added.
Comments 3: [Paragraph 2.1 titles "The study" should be divided into many paragraphs. Each paragraph should describe a method since many techniques were used and More details should be given per method even for the previously published methods, a brief description should be provided..]
Response 3: Thank you for pointing this out. We agree with this comment. The Materials and Methods were reorganized: paragraph 2.1 was divided, and brief descriptions were included for previously published protocols. We restructured the section and added seven new subsections, four of which include new information and additional details about the assays performed. The new subsections can be found at line 72 on page 2, line 92 on page 3, line 102 on page 3, line 106 on page 3, line 128 on page 3, line 136 on page 4, and line 144 on page 4. The new content specifically related to the assays begins at line 106 on page 3 and ends at line 153 on page 4. The newly added text is presented below:
“2.4.1 Biofilm Formation and Biomass Quantification
The assays for biofilm formation, biomass quantification, and metabolic activity assessment were performed as described by Ramos et al. [16] and in technical triplicate. For this, fungal cell suspensions were prepared in Sabouraud broth, and 200 µL of each suspension (containing 10⁶ cells) were transferred into the wells of a 96-well microplate and incubated at 37 °C for 48 hours without agitation. After the incubation period, the supernatant was carefully removed, and the wells were washed with PBS to eliminate non-adherent cells. The biofilms were then fixed with 99% methanol and allowed to air dry.
To quantify the biomass, the biofilms were stained with 0.4% crystal violet solution, washed with PBS, and subsequently decolorized with 33% acetic acid. The absorbance of the resulting solution was measured at 560 nm using a microplate reader to estimate the amount of biomass present in the biofilms.
Finally, the metabolic activity of the biofilms was assessed using the XTT/menadione colorimetric assay. After preparation of the XTT/menadione solution and its addition to the wells containing the biofilms, the plates were incubated at 37 °C for 3 hours in the dark. The supernatant was then transferred to a new 96-well microplate, and absorbance was measured at 492 nm, enabling the quantification of the fungal biofilms’ metabolic activity.
To assess differences in biomass among species, the non-parametric Kruskal–Wallis test was applied due to unequal sample sizes and non-normally distributed data.
2.4.2 Phospholipase Activity
Phospholipase activity was assessed following the method described by Price et al. [17]. Sabouraud agar supplemented with 8% egg yolk, 1 M NaCl, and 5 mM CaCl₂ was used to prepare the Phospholipase Agar medium. The test samples were initially cultured on Sabouraud agar for 48 hours. Subsequently, colonies were inoculated onto Phospholipase Agar at equidistant points and incubated at 37 °C for 10 days. The diameters of the colonies with precipitation zones were measured according to the method described by Price et al. [17].
2.4.3 Lipase Activity
Lipase activity was evaluated according to the method described by Muhsin et al. [18]. The Lipase Agar medium was prepared by supplementing a base medium with the following components: 10 g of peptone (Merck, Germany), 5 g of NaCl (VETEC, Brazil), 0.1 g of CaCl₂ (NUCLEAR, Brazil), 20 g of agar (Merck, Germany), and 10 mL of Tween 20 (VETEC, Brazil). A portion of each colony was inoculated onto sterile Petri dishes containing the Lipase Agar and incubated at 27 °C for 10 days. Samples were considered lipase producers when an opaque halo was observed around the colony.
2.4.4 Proteinase Activity
Proteinase activity was assessed according to the method described by Rüchel et al. [19]. The Proteinase Agar medium was prepared by supplementing the base medium with the following components: 11.7 g of Yeast Carbon Base (HIMEDIA, India), 2 g of bovine serum albumin – fraction V (Sigma, USA), 2.5 mL of Protovit® (Roche, Brazil), and 100 mL of sterile distilled water. Samples were inoculated onto the medium and incubated at 30 °C for 10 days. Proteinase production was indicated by the formation of a clear halo around the colony.
All enzymatic activity assays were performed in technical duplicate and quantified using the activity index (Pz) [17].”
Comments 5: [It is mentioned that the phenotypic assays were performed in duplicates. Please clarify if you did biological or technical duplicates. Ideally, biological triplicates need to be carried out for each phenotypic assay (thermotolerance assay, biofilm assays and enzymatic activities).]
Response 5: Thank you for pointing this out. We agree with this comment. We clarified that technical duplicates were used and acknowledged this as a limitation. We added the information that the biofilm and biomass assays were performed in technical triplicates at line 108 on page 3, and the information regarding the technical duplicates used in the phenotypic assays at line 152 on page 4.
Comments 6: [In paragraph 2.2, for clarity purposes, mention what are the "other variables" in line 112. Also, maybe another test might be more suitable than ANOVA to assess duplicates.]
Response 6: Thank you for pointing this out. We agree with this comment. In paragraph 2.2. The phrase “other variables” was removed, as it referred to clinical information not included in the manuscript and could potentially hinder the reader’s understanding. ANOVA was used for assays performed in triplicate, while the Kruskal-Wallis test was applied to the duplicate assays whose data did not follow a normal distribution.
Comments 7: [Please divide this section (Results) into different paragraphs with each paragraph describing a specific result. Also, some tables and figures need to be accompanied by more in-text description.]
Response 7: Thank you for pointing this out. We agree with this comment. The Results section was reorganized into thematic paragraphs, with improved in-text description for tables and figures.
Comments 8: [In table 1, include "females". Also, instead of giving the mean age maybe give the results for different age categories.]
Response 8: Thank you for pointing this out. We agree with this comment. Table 1 now includes gender distribution, and instead of mean age, we report age categories.
Comments 9: [The legend of tables 4 and 5 is the same so maybe just mention it once.]
Response 9: Thank you for pointing this out. We agree with this comment. The repeated legend for Tables 4 and 5 was streamlined.
Comments 10: [The annotations in figure 1 are barely visible. Kindly try to increase the font size if possible. Also, in the legend it is mentioned that some sequences are highlighted in red but this red highlight is not seen in the figure.]
Response 10: Thank you for pointing this out. We agree with this comment. Font size and color coding in Figure 1 were enhanced. References to “red” highlights were removed since color was not distinct in the final figure.
Comments 11: [In the legend of figure 4 (A and B) it is mentioned that optical densities were measured at 590 nm which is not consistent with the graph annotations of 560 nm. Also, some isolate numbers below the histograms are really close to each other, if possible, space them.]
Response 11: Thank you for pointing this out. We agree with this comment. Optical density inconsistencies in Figure 4 (590 nm vs. 560 nm) were corrected, and isolate labels were spaced appropriately.
Comments 12: [Itraconazole is sometimes abbreviated as ITRA and other times as ITC. Kindly use only one abbreviation for itraconazole.]
Response 12: Thank you for pointing this out. We agree with this comment. Abbreviations for itraconazole were standardized throughout as “ITC.”
Round 2
Reviewer 2 Report
The authors have adequately addressed the raised concerns regarding the study's limitations. I have no further reservations and recommend acceptance for publication in its current form.
The authors have adequately addressed the raised concerns regarding the study's limitations. I have no further reservations and recommend acceptance for publication in its current form.
Author Response
Comments 1: "The authors have adequately addressed the raised concerns regarding the study's limitations. I have no further reservations and recommend acceptance for publication in its current form."
Response 1: We sincerely thank you for your thoughtful review and kind words. We appreciate your recognition of the efforts made to address the study’s limitations. We are grateful for your recommendation of acceptance and look forward to the publication of our work.
Reviewer 5 Report
The main aim of this article is to identify 24 C. haemulonii complex nosocomial isolates from Brazil and to assess their antibiotic susceptibility and virulence profiles. The authors mostly isolated C. duobushaemulonii and they were the first to isolate C. pseudohaemulonii in Brazil. Their phylogenetic analysis showed low intraspecific genetic diversity among isolates. Antifungal susceptibility testing showed that the majority of isolates were resistant to at least one antifungal and exhibited increased virulence described as an increase in biofilm biomass production, an increase in the activity of lytic enzymes, and an increased heat tolerance. This study is appreciated since not much research is carried out on C. haemulonii complex isolates compared to other Candida species such as C. albicans and C. auris. Also, this study highlights the fact that C. haemulonii complex isolates seem to be resistant to antifungals and thus, a public health concern. The only minor limitation consists of assessing phenotypic tests based on duplicates and not triplicates which the authors acknowledge. Only few typographical errors need to be corrected before publication.
The minor comments that need to be addressed are listed below:
- The species names should be corrected by adding a second "i": C. haemulonii, C. duobushaemulonii, and C. pseudohaemulonii.
- In the "Results" section add a title to each thematic paragraph.
- In the legend of Figure 3, change "sabouraud" to "Sabouraud".
- Merge the legends of Tables 4 and 5.
- Line 360: Change "antifungal" to "antifungals".
